# MSCs rescue impaired wound healing in a murine LAD1 model by adaptive responses to low TGF-β1 levels

Dongsheng Jiang[†] (iD), Karmveer Singh (iD), Jana Muschhammer, Susanne Schatz, Anca Sindrilaru, Evgenia Makrantonaki, Yu Qi, Meinhard Wlaschek & Karin Scharffetter-Kochanek[*] (iD)

## Abstract

Mutations in the CD18 gene encoding the common β-chain of β2 integrins result in impaired wound healing in humans and mice suffering from leukocyte adhesion deficiency syndrome type 1 (LAD1). Transplantation of adipose tissue-derived mesenchymal stem cells (MSCs) restores normal healing of CD18$^{-/-}$ wounds by restoring the decreased TGF-β1 concentrations. TGF-β1 released from MSCs leads to enhanced myofibroblast differentiation, wound contraction, and vessel formation. We uncover that MSCs are equipped with a sensing mechanism for TGF-β1 concentrations at wound sites. Low TGF-β1 concentrations as occurring in CD18$^{-/-}$ wounds induce TGF-β1 release from MSCs, whereas high TGF-β1 concentrations suppress TGF-β1 production. This regulation depends on TGF-β receptor sensing and is relayed to microRNA-21 (miR-21), which subsequently suppresses the translation of Smad7, the negative regulator of TGF-β1 signaling. Inactivation of TGF-β receptor, or overexpression or silencing of miR-21 or Smad7, abrogates TGF-β1 sensing, and thus prevents the adaptive MSC responses required for tissue repair.

**Keywords** chronic wounds; environment sensing; LAD1; mesenchymal stem cells; microRNA-21
**Subject Categories** Molecular Biology of Disease; Signal Transduction; Skin

## Introduction

Mutations in the gene encoding CD18 (β2 integrin, Itgb2), the common β-chain of the β2 integrin family, result in severe wound healing disturbances in human patients with leukocyte adhesion deficiency syndrome type 1 (LAD1) [1,2]. Previously, we have generated a CD18-deficient (CD18$^{-/-}$) mouse model, which closely mirrors human LAD1 with significantly reduced healing of full-thickness wounds [3,4]. Due to CD18 deficiency and the lack of functional β2 integrins, wound macrophages were not able to adhere to neutrophils and thus could not phagocytose neutrophils. As neutrophil phagocytosis by macrophages is important to activate the release of transforming growth factor beta 1 (TGF-β1) from macrophages [5], CD18 deficiency led to disrupted phagocytosis of neutrophils and reduced TGF-β1 release from macrophages at the wound site [3]. TGF-β1 is critical to terminate the inflammatory phase of wound healing and to initiate the phase of granulation tissue formation, wound contraction, and angiogenesis essentially required for sufficient supply with oxygen and nutrients. As expected, decreased TGF-β1 concentrations in CD18$^{-/-}$ wounds result in decreased myofibroblast differentiation, reduced wound contraction, and severely reduced angiogenesis, which finally leads to delayed wound repair [3]. Due to the lack of wound contraction and delayed healing, LAD1 patients very often suffer from life-threatening infections [3]. TGF-β1 is essential for the differentiation of alpha-smooth muscle actin (α-SMA)-positive myofibroblasts, which are the key for wound contraction [6]. Previously, we showed that injection of recombinant TGF-β1 restored wound healing in the CD18-deficient murine LAD1 model [3]; however, in human LAD1 patients as well as in patients with TGF-β1-deficient chronic wounds, it is much more complicated as recombinant growth factors like TGF-β1 due to enhanced proteolytic activity at the wound site are easily degraded in chronic human wounds [7–9]. In consequence, the treatment with TGF-β1 injections at the wound site has not been successfully established in clinical routine.

In recent decades, mesenchymal stem cells (MSCs) have been intensively studied for their potential for cell-based therapy and regenerative medicine [10,11]. These multipotent cells are found among mature cells in different tissues and organs, and exert beneficial effects for tissue repair through several mechanisms, such as resolving inflammation and the release of paracrine factors, thus acting as effective "drug store" [12]. We and others have shown that MSCs accelerate healing in acute wounds [13,14], and in addition reduce scar formation [14,15].

---

Department of Dermatology and Allergic Diseases, University of Ulm, Ulm, Germany
*Corresponding author. Tel: +49 731 50057501; Fax: +49 731 50057502; E-mail: karin.scharffetter-kochanek@uniklinik-ulm.de
†Present address: Comprehensive Pneumology Center, Institute of Lung Biology and Disease, Helmholtz Zentrum München, Munich, Germany

Interestingly, earlier studies suggested that certain environmental stimuli could change the properties of MSCs. In this regard, mechanical loading induces matrix metalloprotease 2, the release of TGF-β1, and basic fibroblast growth factor from MSCs, and in consequence, enhanced angiogenesis was observed [16]. Immuno-suppressive properties of MSCs depend on Toll-like receptor-specific priming [17] and the specific cytokine environment [18]. However, sensing of their tissue neighborhood and adaptive responses that MSCs may potentially mount is largely unexplored. Inspired by these data and the urgent need to develop strategies to treat chronic wounds according to their changing microenvironment, we wished to test the hypothesis that MSCs sense their neighborhood and through a currently unknown signaling cascade mount an adaptive response in the interest of tissue repair and maintenance.

In this study, we investigated the effect of adipose tissue-derived MSCs (AT-MSCs) on LAD1 chronic wounds employing the full-thickness excisional wound model in CD18$^{-/-}$ mice and explored the underlying mechanisms. We found that MSCs actively sense their microenvironment at the wound site and respond accordingly in the overall interest of accelerated tissue repair. This suggests a so far underestimated advantage of MSC-based therapy over the application of recombinant proteins to treat LAD1 chronic wounds or other non-healing wounds.

# Results

### MSCs accelerate healing of CD18$^{-/-}$ wounds

In order to investigate whether AT-MSCs have beneficial effects on the healing of CD18$^{-/-}$ wounds, four full-thickness excisional wounds were produced on the dorsal skin of CD18$^{-/-}$ and WT mice. $1 \times 10^6$ AT-MSCs were intradermally injected into wound margins 1 day post-wounding. The groups received PBS injections served as controls. As we reported previously [3], the CD18$^{-/-}$ wounds received PBS injections showed delayed healing compared to the PBS-injected WT mice, illustrated by larger wound areas measured at days 3, 5, 7, and 9 post-wounding (Fig 1A and B). More importantly, injection of AT-MSCs resulted in significant acceleration of healing in CD18$^{-/-}$ wounds at all studied time points, compared to PBS-injected CD18$^{-/-}$ wounds. The healing kinetics was rescued to that in WT wounds, as indicated by similar wound sizes of AT-MSC-injected CD18$^{-/-}$ wounds and PBS-injected WT wounds (Fig 1A and B). Notably, AT-MSCs also accelerated healing of WT wounds, in line with our previous studies [13,14].

### MSC-derived TGF-β1 is responsible for myofibroblast differentiation and restoration of impaired wound healing in CD18$^{-/-}$ mice

To assess whether injected AT-MSCs were present in the wounds, the wound sections were immunostained with an antibody against human beta-2 microglobulin (β2M) that specifically detects human AT-MSCs at the wound site of murine wounds. AT-MSCs were found in the wound margins of both WT and CD18$^{-/-}$ wounds at day 2 post-wounding, and the number of cells decreased at day 5 (Fig 2A). We further quantified the number of transplanted MSCs in WT and CD18$^{-/-}$ wounds by quantification of the amount of

human-specific *Alu* DNA in wounds at days 2, 5, and 7 post-wounding (Fig 2B). The transplanted MSCs are lost gradually in wounds, with an improved survival of injected MSCs in CD18$^{-/-}$ wounds as opposed to WT wounds (Fig 2B). The longer survival of transplanted MSCs in CD18$^{-/-}$ mice is likely due to disrupted binding of CD18$^{-/-}$ NK cells to xenograft MSCs [19]. These data suggest that accelerated healing was rather due to paracrine effects and less likely to MSC differentiation and replacement of the injured tissue.

Previously, we found significantly less TGF-β1 in CD18$^{-/-}$ wounds, which resulted in impaired formation of granulation tissue at days 5–7 post-wounding [3]. TGF-β1 concentrations were measured in the tissue lysates from CD18$^{-/-}$ and WT wounds that were injected with either AT-MSCs or PBS. As early as day 2 post-wounding (day 1 post-injection), AT-MSC-injected CD18$^{-/-}$ wounds showed significantly higher TGF-β1 concentrations compared to PBS-injected wounds. At day 5 post-wounding, as expected, PBS-injected CD18$^{-/-}$ wounds showed 50% less TGF-β1 compared to PBS-injected WT wounds. Of note, we observed a substantial increase in TGF-β1 concentrations in AT-MSC-injected CD18$^{-/-}$ wounds, which were fully restored to physiologic TGF-β1 concentrations of WT wounds (Fig 2C). These data suggest that restoration of the TGF-β1 concentrations is causal for accelerated healing of AT-MSC-injected CD18$^{-/-}$ wounds.

Since the TGF-β1 ELISA could not distinguish between mouse and human origin, primers that allow for specific amplification of human TGF-β1 as opposed to murine TGF-β1 were employed. Interestingly, both mouse and human TGF-β1 mRNA were detected in AT-MSC-injected wounds at day 2 post-wounding (Fig EV1A). The expression of murine TGF-β1 mRNA levels was not significantly different in CD18$^{-/-}$ wounds at days 2, 5, and 7 post-wounding (Fig EV1B). By contrast, human TGF-β1 expression normalized on the numbers of MSCs in wounds showed a distinctive pattern in WT as opposed to CD18$^{-/-}$ wounds. In WT wounds, human TGF-β1 mRNA expression per MSC was upregulated at day 2, decreased at day 5, and was not detected at day 7 post-wounding (Fig 2D). This time course is in line with our previous observation on bone marrow-derived MSCs in acute wound healing [14]. By contrast, human TGF-β1 expression per MSC in CD18$^{-/-}$ wounds kept upregulated throughout the healing process and was found at high level at day 7 (Fig 2D). These data suggest that the increase in TGF-β1 concentrations in AT-MSC-injected CD18$^{-/-}$ wounds was mainly contributed by human TGF-β1 released from the injected AT-MSCs eventually resulting in restoration of TGF-β1 levels in CD18$^{-/-}$ wounds at day 5 post-wounding.

To verify whether increased TGF-β1 concentration in AT-MSC-transplanted CD18$^{-/-}$ wounds promoted granulation tissue formation, cryosections of AT-MSC- or PBS-injected CD18$^{-/-}$ wounds were immunostained for the myofibroblast marker α-SMA. At day 5 post-wounding, injection of AT-MSCs resulted in higher α-SMA expression in CD18$^{-/-}$ wounds compared to PBS-injected wounds, and this was even more prominent in day 7 wounds (Fig 3A and B). Costaining for human-specific β2M and TGF-β1, or β2M and α-SMA on cryosections of day 5 AT-MSC-injected CD18$^{-/-}$ wounds depicted β2M to be co-localized with TGF-β1 but not with α-SMA (Fig 3C). These data indicate that AT-MSCs, while enforcing granulation tissue formation by paracrine release of TGF-β1, did not differentiate to myofibroblasts themselves.

To further confirm that TGF-β1 released by human AT-MSCs is responsible for wound fibroblasts to myofibroblast differentiation in

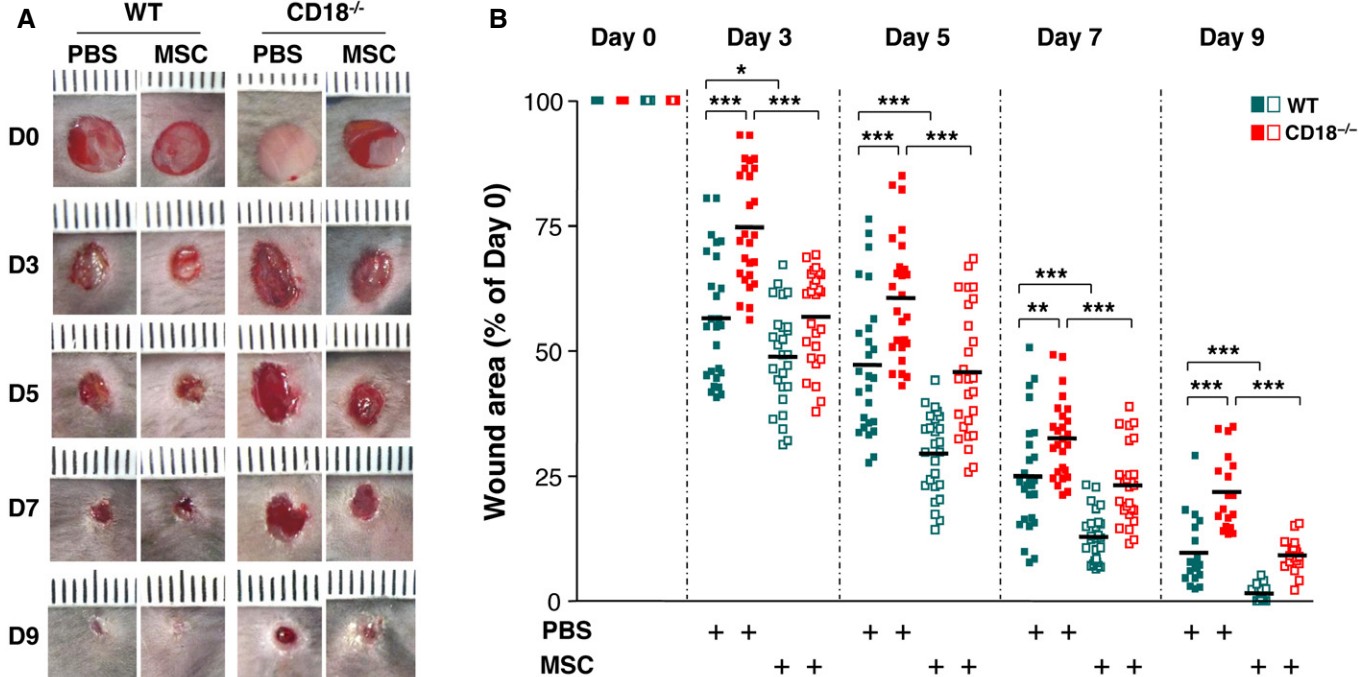

**Figure 1.  MSCs accelerate healing of CD18$^{-/-}$ wounds.**

A  Four full-thickness excisional wounds were produced on each of CD18$^{-/-}$ or WT mice by 6-mm biopsy punches. Each wound received intradermal injection of 2.5 × 10$^5$ AT-MSCs or PBS control. Each wound region was digitally photographed at the indicated time points, and wound areas were analyzed using Adobe Photoshop. Depicted are representative macroscopic pictures of PBS-injected or AT-MSC-injected WT or CD18$^{-/-}$ wounds at days 0, 3, 5, 7, and 9 post-wounding.

B  Quantitative analysis of 20 wound areas per group, expressed as percentage of the initial wound size at day 0. The line in each group represents the mean value of 20 wounds from five mice. *$P < 0.05$, **$P < 0.01$, ***$P < 0.001$ by Student's *t*-test with Welch's correction.

Source data are available online for this figure.

CD18$^{-/-}$ wounds, AT-MSCs were transfected with TGF-β1 siRNA with 4 different sequences or control siRNAs. The siRNA with the best TGF-β1 silencing effect was selected for further experiments. This TGF-β1 siRNA showed more than 90% suppression at both mRNA and protein levels (Fig EV1C). Human AT-MSCs were cocultured with murine primary dermal fibroblasts for 48 h with the ratio of fibroblasts:MSCs at 10:1. Cultures were immunostained with human-specific β2M in green and α-SMA in red. AT-MSCs did not show α-SMA expression (Fig 4A and G). Murine fibroblasts at early passages displayed only low basal α-SMA expression (Fig 4B and G). By contrast, after treatment with 2 ng/ml of recombinant human TGF-β1, murine dermal fibroblasts expressed large amount of α-SMA and showed clear actin bundles (Fig 4C and G). This indicates that human TGF-β1 was fully functional and induced myofibroblast differentiation in murine fibroblasts. This observation was further supported by Western blot analysis from cell lysates either of murine fibroblasts cocultured with human AT-MSCs or from lysates of murine fibroblasts treated with recombinant human TGF-β1. Under these conditions, a strong expression of α-SMA (Fig EV2A) and phosphorylated Smad2, a downstream transcription factor of TGF-β1 signaling, was observed (Fig EV2B). Coculture of murine fibroblasts and AT-MSCs revealed high α-SMA expression in murine fibroblasts (Fig 4D and G), often in the close proximity with AT-MSCs, but not in AT-MSCs (Fig 4D), in line with a recent report that MSCs released TGF-β1-induced α-SMA expression in cocultured dermal fibroblasts [20]. The fibroblasts expressed only basal level of

α-SMA when cocultured with TGF-β1-silenced AT-MSCs (Fig 4E and G), but showed high α-SMA expression in the coculture with control siRNA-transfected AT-MSCs (Fig 4F and G). These data confirmed our *in situ* findings that human TGF-β1 released by AT-MSCs induced myofibroblast differentiation in CD18$^{-/-}$ wounds (Fig 3C).

To investigate whether TGF-β1 is responsible for the beneficial effect of AT-MSCs on the impaired healing of CD18$^{-/-}$ wounds, TGF-β1-silenced AT-MSCs or control siRNA-transfected AT-MSCs were intradermally injected into CD18$^{-/-}$ wounds. Untransfected AT-MSCs served as positive control, and human dermal fibroblasts (HDFs) or PBS served as negative controls. Silencing of TGF-β1 in AT-MSCs did not affect transplantation efficiency, survival, or proliferation of MSCs in wounds (Fig EV3). In line with the result shown in Fig 1, AT-MSCs markedly accelerate healing of CD18$^{-/-}$ wounds compared to wounds injected with HDFs or PBS at all studied time points. Of note, wounds injected with TGF-β1-silenced AT-MSCs were significantly larger compared to those injected with AT-MSC- or control siRNA-transfected AT-MSCs at days 3, 5, 7, and 10 post-wounding (Fig 4H).

To explore whether the lack of healing in CD18$^{-/-}$ wounds when injected with TGF-β1-silenced MSCs, in fact, is due to reduced TGF-β1 at the wound site, TGF-β1 mRNA and protein concentrations were assessed in wound tissue. Compared to high TGF-β1 mRNA levels in CD18$^{-/-}$ wounds injected with control siRNA-transfected MSCs, wounds injected with TGF-β1-silenced MSCs revealed significantly lower expression of human TGF-β1 mRNA at day 2

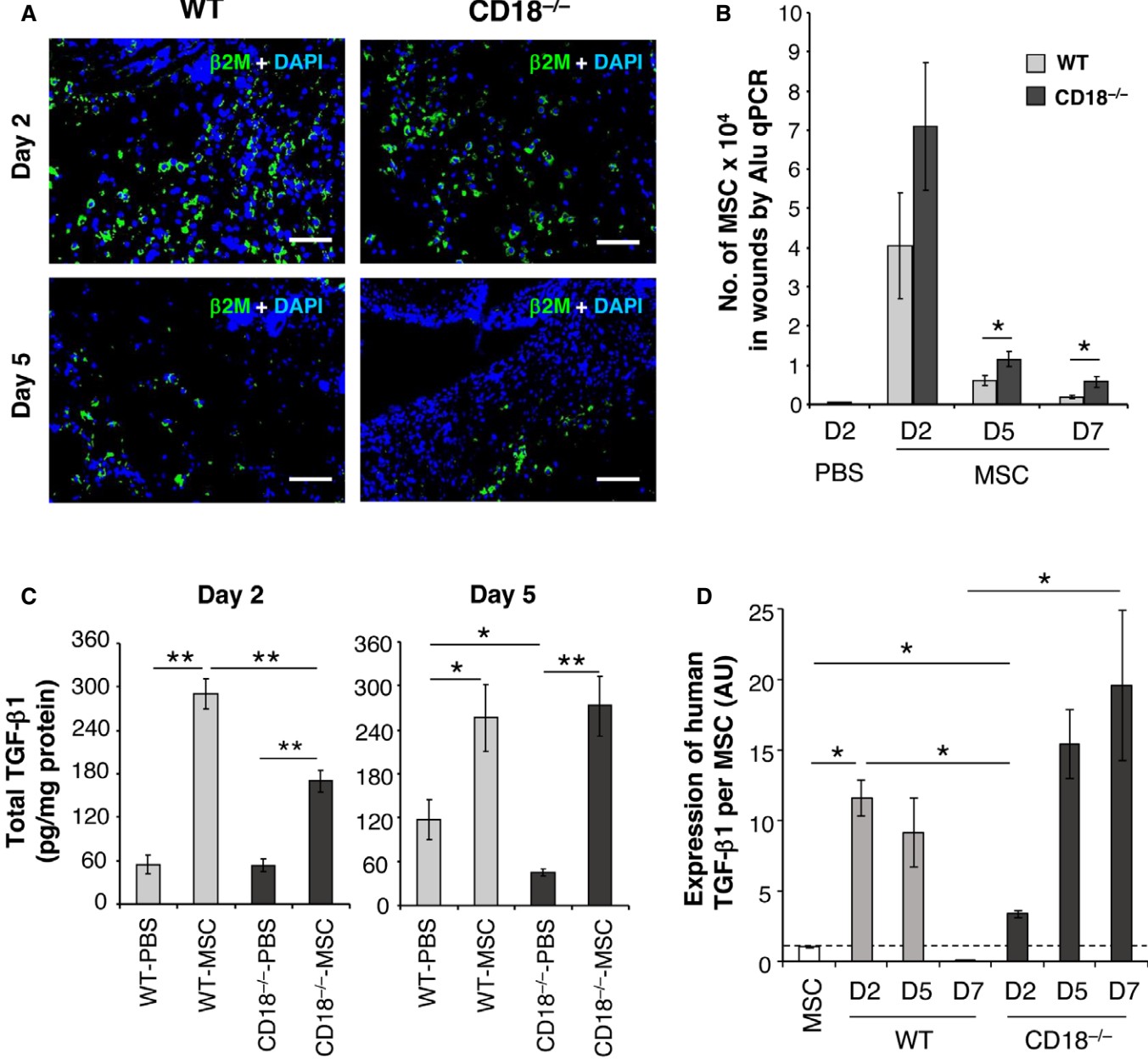

**Figure 2. Injection of MSCs restores TGF-β1 concentration in CD18⁻/⁻ wounds.**

A   AT-MSCs or PBS was intradermally injected to WT or CD18⁻/⁻ wounds, and the wound tissues were harvested at days 2, 5, and 7 post-wounding. Depicted are representative photographs of wound cryosections at days 2 and 5 post-wounding stained for human-specific β2M (green). Nuclei were stained with DAPI (blue). Scale bars: 100 μm.

B   Genomic DNA was isolated from the wound tissues, and the amount of human-specific *Alu* DNA was quantified by real-time PCR and then converted to the number of AT-MSCs based on a standard curve. Data are expressed as mean ± SEM, *n* = 3 at each time point, *P < 0.05 by two-tailed unpaired *t*-test.

C   Total TGF-β1 concentrations in wound lysates were measured by TGF-β1-specific ELISA. Results are expressed as mean ± SEM, *n* = 3 at each time point, *P < 0.05, **P < 0.01 by one-way ANOVA with Tukey's test.

D   The expression of human TGF-β1 in cultured AT-MSCs and wound tissues of WT or CD18⁻/⁻ wounds received AT-MSCs injection was quantified by qPCR with primers specifically amplify human TGF-β1, and normalized on the numbers of MSCs in each condition shown in (B). Data are expressed as mean ± SEM, *n* = 8 wounds (two wounds × four mice) per time point, *P < 0.05 by one-way ANOVA with Tukey's test. AU, arbitrary unit.

Source data are available online for this figure.

post-wounding (Fig 5A). This corresponds to significantly reduced concentrations of total TGF-β1 protein (26.3% reduction; Fig 5B) and active TGF-β1 protein (37.6% reduction; Fig 5C) at day 5, as well as to lower expression of α-SMA (Fig 5D and E) and CD31 (Fig 5F and G) in wound margins at days 5 and 7. Notably, compared to dermal fibroblasts, TGF-β1-silenced AT-MSCs still

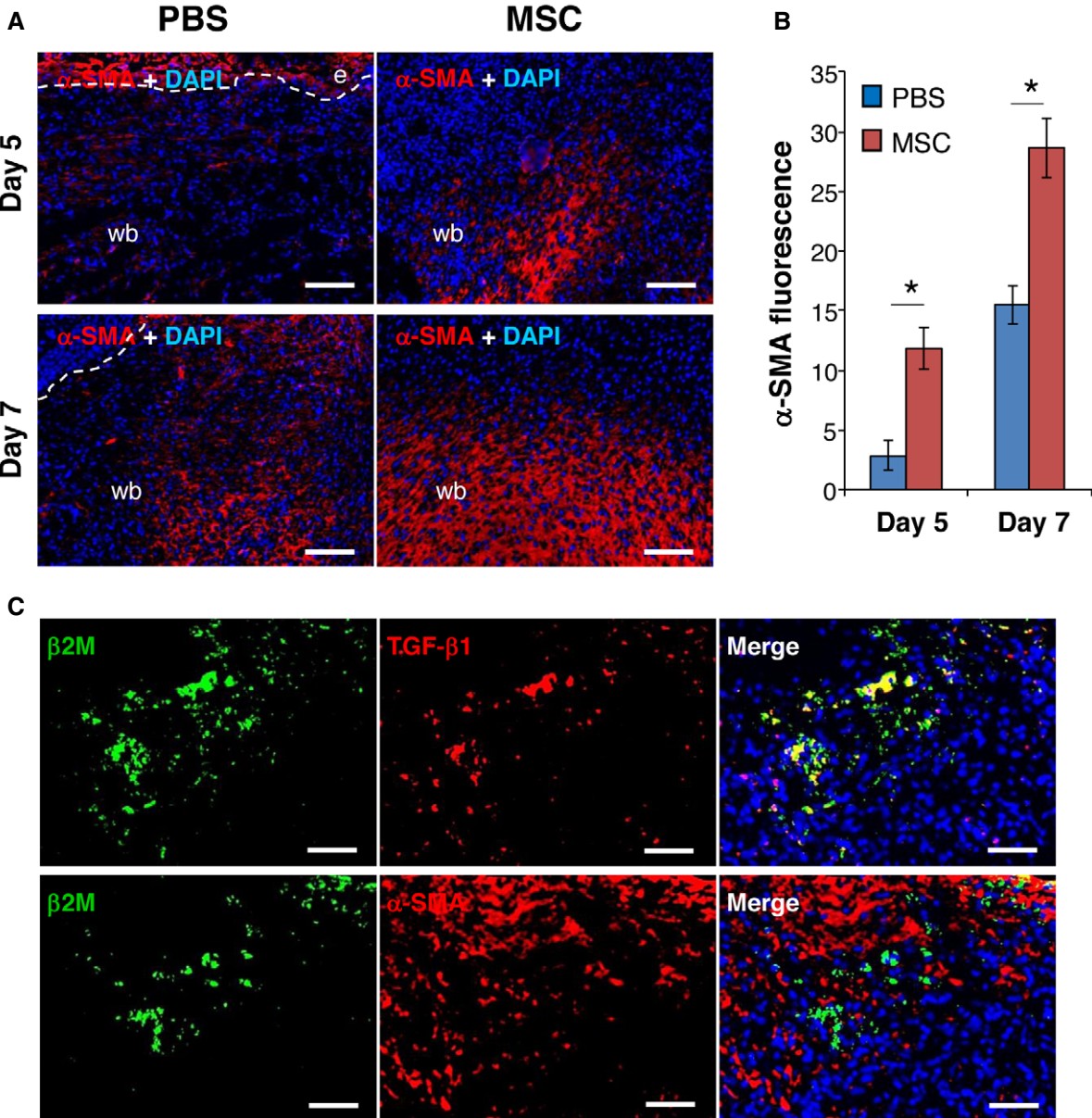

**Figure 3. Injection of MSCs induces granulation tissue formation in CD18$^{-/-}$ wounds.**

A  Cryosections of PBS- or AT-MSC-injected CD18$^{-/-}$ wounds at days 5 and 7 post-wounding were immunostained for α-SMA (red). Nuclei were stained with DAPI (blue). The dashed lines separate epidermis and wound bed. e, epidermis; wb, wound bed. Scale bars: 100 μm.

B  Quantification of α-SMA immunofluorescence shown in (A), mean ± SEM, *n* = 3 wounds per group, *P < 0.05 by two-tailed unpaired *t*-test.

C  Cryosections of AT-MSC-injected CD18$^{-/-}$ wounds at day 5 post-wounding were co-immunostained for human β2M (green) and TGF-β1 (red, upper panel) or α-SMA (red, lower panel). The depicted results are representative photographs of three independent experiments. Scale bars: 50 μm.

Source data are available online for this figure.

showed beneficial effect on CD18$^{-/-}$ wound healing from day 5 onwards (Fig 4H). This indicates that the released TGF-β1 was partially but not exclusively responsible for the restoration of impaired healing in AT-MSC-injected CD18$^{-/-}$ wounds. Together, these data demonstrate that AT-MSCs constitute the major source of TGF-β1, which in consequence induced differentiation of murine wound-resident fibroblasts to myofibroblast, and promoted granulation tissue formation and wound contraction in CD18$^{-/-}$ wounds.

**Adaptive TGF-β1 release from MSCs is regulated by miR-21/ Smad7 and depends on sensing of environmental TGF-β1 levels at the wound site**

The release of lower TGF-β1 concentrations from injected AT-MSCs in WT and high TGF-β1 concentrations in CD18$^{-/-}$ wounds suggests that AT-MSCs are endowed with a mechanism of sensing TGF-β1 concentrations at the wound site and, in consequence,

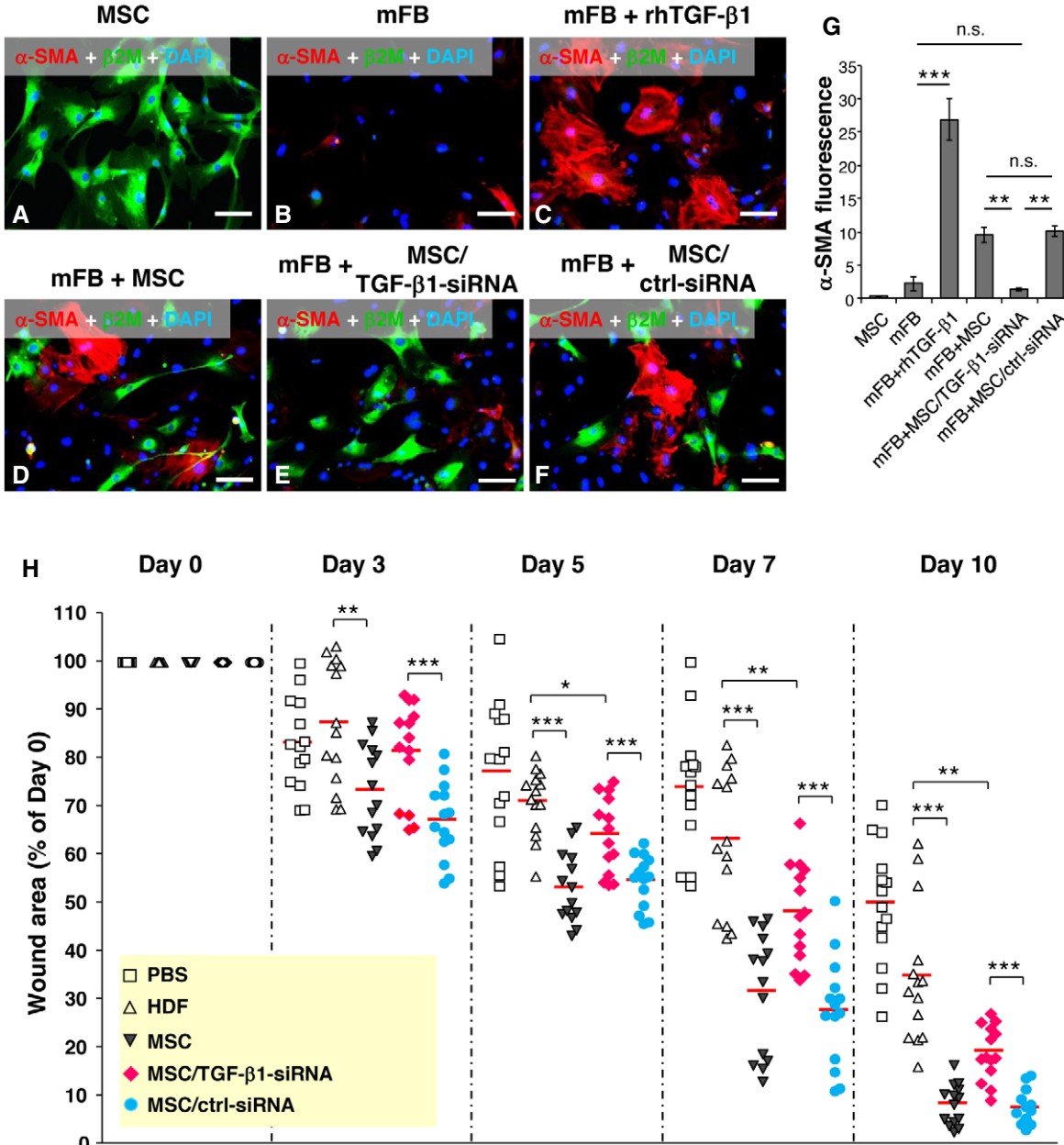

**Figure 4. TGF-β1 released by MSCs is responsible for accelerated healing in CD18$^{-/-}$ wounds.**

A–F Representative photographs of immunofluorescence staining for α-SMA (red) and human β2M (green) on cultured human AT-MSCs (A), murine primary dermal fibroblasts (mFB) (B), mFB treated with 2 ng/ml recombinant human TGF-β1 (C), and cocultures of mFB with AT-MSCs (D) or TGF-β1 siRNA-transfected AT-MSCs (E) or control siRNA-transfected AT-MSCs (F) after 48-h culture. Nuclei were counterstained with DAPI (blue). Scale bars: 100 μm.

G Quantification of α-SMA immunofluorescence shown in (A–F), mean ± SEM, n ≥ 3 wells per group, **P < 0.01; ***P < 0.001; n.s., not significant, by one-way ANOVA with Tukey's test.

H Four full-thickness excisional wounds were produced on each of CD18$^{-/-}$ mouse by 6-mm biopsy punches. Each wound received intradermal injection of 2.5 × 10$^5$ AT-MSCs or human dermal fibroblasts (HDFs) or TGF-β1 siRNA-transfected AT-MSCs or control siRNA-transfected AT-MSCs or PBS. Each wound region was digitally photographed at days 0, 3, 5, 7, and 10 post-wounding, and wound areas were analyzed using Adobe Photoshop. The depicted result is the quantitative analysis of all wound areas per group, expressed as percentage of the initial wound size at day 0. The line in each group represents the mean value. *P < 0.05, **P < 0.01, ***P < 0.001 by two-tailed unpaired t-test with Welch's correction.

Source data are available online for this figure.

adaptively shape an effector response with the release of different TGF-β1 concentrations. To test this hypothesis, AT-MSCs were exposed to different concentrations of recombinant human TGF-β1

in culture media containing [$^{35}$S]-Met/Cys overnight. The newly synthesized radioactive TGF-β1 was captured in supernatants with anti-TGF-β1 antibody. Exogenously added TGF-β1 at concentrations

from 0 to 0.3 ng/ml induced the release of TGF-β1 from AT-MSCs in a dose-dependent manner. By contrast, higher TGF-β1 concentrations (1–10 ng/ml) in the culture medium significantly suppressed

TGF-β1 release from AT-MSCs (Fig 6A). The supernatants were subjected to a second round of TGF-β1 capture revealing only background noise. This finding reliably ruled out the possibility that the

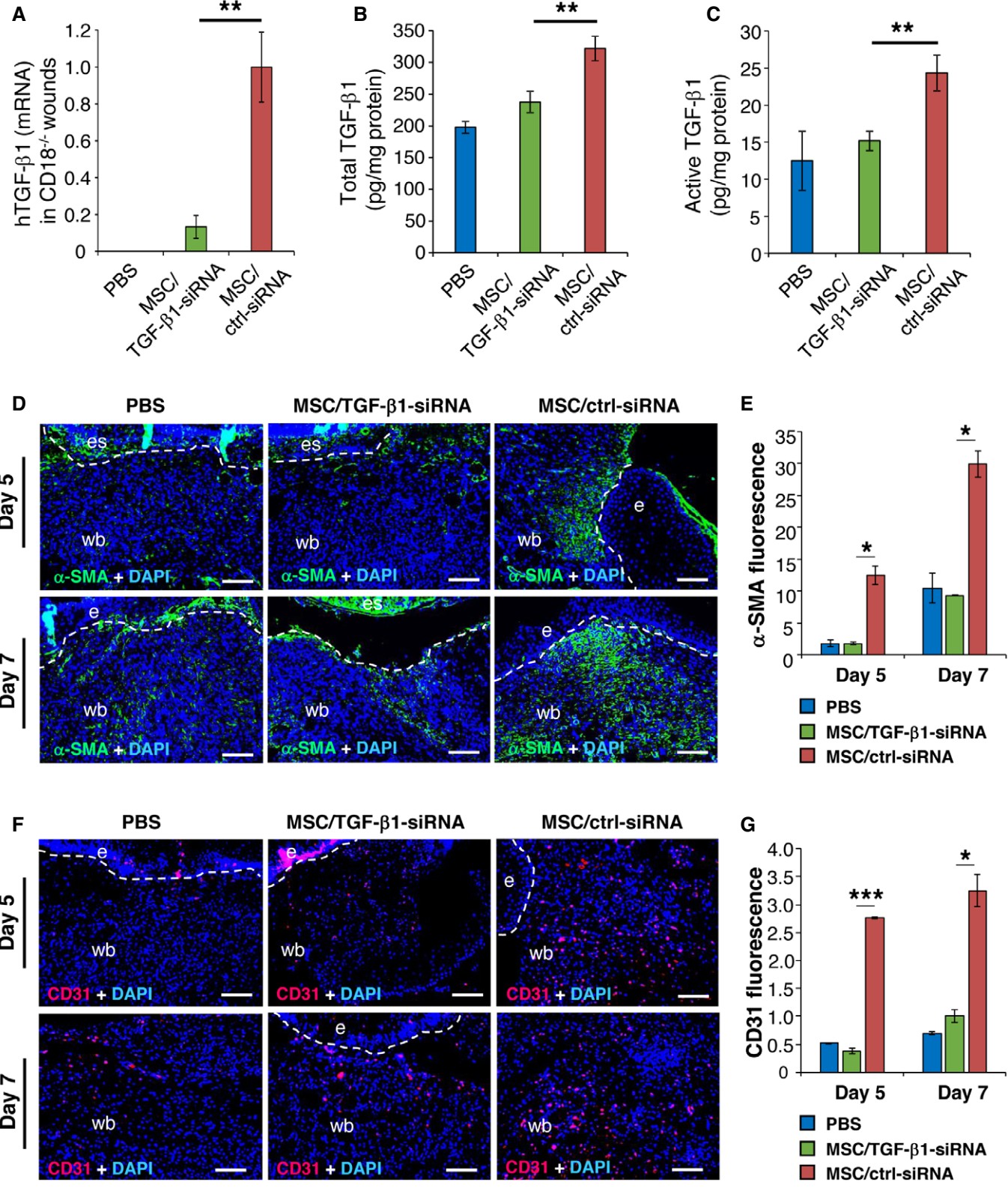

Figure 5.

◀

Figure 5. TGF-β1 released by MSCs contributes to induced myofibroblast differentiation and granulation tissue formation.

A–C   2.5 × 10$^5$ of TGF-β1 siRNA or control siRNA-transfected AT-MSCs were intradermally injected around each of CD18$^{-/-}$ murine wound. PBS mock injection served as negative control. Wound tissue was harvested at day 2, 5, and 7 post-wounding for quantification of human TGF-β1 mRNA (A) at day 2 by qPCR, total TGF-β1 (B), and active TGF-β1 (C) protein at day 5 by ELISA. Data are expressed as mean ± SEM, n = 3 wounds per group, **$P < 0.01$, by one-way ANOVA with Tukey's test.

D–G   Expression of α-SMA (D and E) and CD31 (F and G) at days 5 and 7 by immunostaining on tissue sections. The dashed lines indicate the border of the wound bed and epidermis or eschar. e, epidermis; es, eschar; wb, wound bed. Scale bars: 100 μm. Quantification data are expressed as mean ± SEM, n = 3 wounds per group, *$P < 0.05$, ***$P < 0.001$, by one-way ANOVA with Tukey's test.

Source data are available online for this figure.

observed decreased TGF-β1 release from AT-MSCs in the presence of higher concentrations of exogenous recombinant TGF-β1 was due to insufficient capture by the TGF-β1-specific antibody.

To investigate the underlying molecular mechanism of the adaptive response of MSCs to environmental TGF-β1 concentrations at the wound site, the expression of Smad7, the negative regulator of TGF-β1 signaling, was explored in AT-MSCs and HDFs exposed to increasing concentrations of recombinant TGF-β1. Smad7 expression at mRNA level was not significantly different in AT-MSCs and HDFs (Fig EV4). However, at protein level high environmental TGF-β1 concentrations (1–10 ng/ml) resulted in upregulation of Smad7 protein in AT-MSCs (Fig 6B) with reduced TGF-β1 release (Fig 6A). These data thus uncover Smad7 as a negative regulator of TGF-β1 release in response to high TGF-β1 concentrations. By contrast, the regulation of Smad7 protein expression in HDFs in response to high environmental TGF-β1 concentrations was different from that in AT-MSCs. HDFs exposed to 3 ng/ml TGF-β1 expressed significantly less Smad7 protein compared to HDFs treated with 1 ng/ml TGF-β1 (Fig 6C). This suggests that Smad7 was differentially regulated in AT-MSCs and HDFs at post-transcriptional level upon exposure to high concentrations of environmental TGF-β1.

To further identify potential candidates that post-transcriptionally regulate Smad7 expression in MSCs in response to environmental TGF-β1, the expression of the most abundantly expressed microRNAs (miRNA) was screened in MSCs treated with 0.1, 1, or 10 ng/ml recombinant TGF-β1 for 24 h and compared to non-treated MSCs (Table EV1). The expression of miRNAs that were up- or down-regulated more than twofold following treatment with recombinant TGF-β1 was summarized in Table 1. Interestingly, microRNA 21 (miR-21) was at the top of the list, with a 5.82-fold upregulation upon exposure to 1 ng/ml of recombinant TGF-β1, with a reduction to 1.16-fold when MSCs were exposed to TGF-β1 at 10 ng/ml. As miR-21 was reported to suppress Smad7 translation [21–24], and Smad7 inhibits TGF-β1 signaling [25–27], we here set out to correlate miR-21 concentrations and Smad7 expression both on mRNA and protein levels in response to increasing TGF-β1 concentrations in the culture medium. We, in fact, found that low TGF-β1 concentrations in culture media (0–1 ng/ml) resulted in an increase in miR-21 expression (Fig 6D), reduced Smad7 concentrations (Fig 6B), and with enhanced release of TGF-β1 from MSCs (Fig 6A). By contrast, higher TGF-β1 concentrations (1–10 ng/ml) in the culture medium led to reduced miR-21 expression (Fig 6D), to higher Smad7 concentrations (Fig 6B), and in consequence, to a significantly reduced TGF-β1 release from MSCs (Fig 6A). By contrast, the expression of miR-21 steadily increased in response to increasing TGF-β1 concentrations in the culture medium up to a maximal

plateau in HDFs (Fig 6D); however, in the presence of high TGF-β1 concentrations in the culture medium, HDFs did not mount any adaptive response. These data strongly suggest that sensing of environmental TGF-β1 levels and subsequent shaping an adapted TGF-β1 release represent a unique property of MSCs but not of dermal fibroblasts. We further hypothesized that sensing environmental TGF-β1 concentrations at the wound site would be initiated by the TGFβ receptor (TGF-βR) on MSCs with subsequent relay of this information to miR-21 and Smad7 signaling which constitute the effector mechanism adaptively shaping the amount of TGF-β1 release from MSCs.

To validate the components of the proposed TGF-β1 sensing and downstream signaling for their causal contribution to the adaptive response, we employed an inhibitor of the TGF-β receptor phosphorylation (activation) and either silenced or overexpressed downstream components of the TGF-β1 signaling in MSCs. In the presence of increasing TGF-β1 concentrations in the culture medium, a steep increase in miR-21 expression was observed up to TGF-β1 concentrations of 1 ng/ml with a sharp decrease in miR-21 expression at higher TGF-β1 concentrations (3–10 ng/ml). By contrast, this bell-shaped curve of miR-21 expression in the presence of increasing TGF-β1 concentrations was completely disrupted when MSCs were pretreated with SB431542, a selective and potent inhibitor of the TGF-βR activating receptor-like kinase (ALK-4,5,7) [28–30] (Fig 7A). In the presence of increasing TGF-β1 concentrations in the culture medium, Smad7-silenced MSCs (Fig 7B and C) displayed a substantially increase in specific TGF-β1 mRNA expression (Fig 7D). These data support the notion that Smad7 is an important downstream effector of environmental TGF-β1 sensing, and at the same time is crucial for shaping the adaptive TGF-β1 release from MSCs. By contrast, under identical conditions, untransfected and control siRNA-transfected MSCs revealed a bell-shaped curve of TGF-β1 mRNA expression with an increase in TGF-β1 mRNA at low TGF-β1 concentrations in the culture medium and a decrease in TGF-β1 mRNA expression in MSCs at higher TGF-β1 concentrations in the culture medium (Fig 7D). Overexpression or silencing miR-21 was achieved by transfecting MSCs with a miR-21 mimic or inhibitor, respectively. As expected, miR-21 mimic-transfected MSCs showed consistent higher miR-21 expression (Fig 7E) and low Smad7 expression (Fig 7F) compared to control mimic-transfected cells. *Vice versa* miR-21 inhibitor-transfected MSCs expressed minimal miR-21 (Fig 7H) and significantly higher Smad7 (Fig 7I) compared to control inhibitor-transfected MSCs. More importantly, disturbing the miR-21 expression pattern in MSCs exposed to increasing concentrations of environmental TGF-β1 by either miR-21 mimic or miR-21 inhibitor, significantly changed the TGF-β1 expression pattern in MSCs (Fig 7G and J). As expected, TGF-β1 mRNA expression was lower in miR-21 inhibitor-transfected

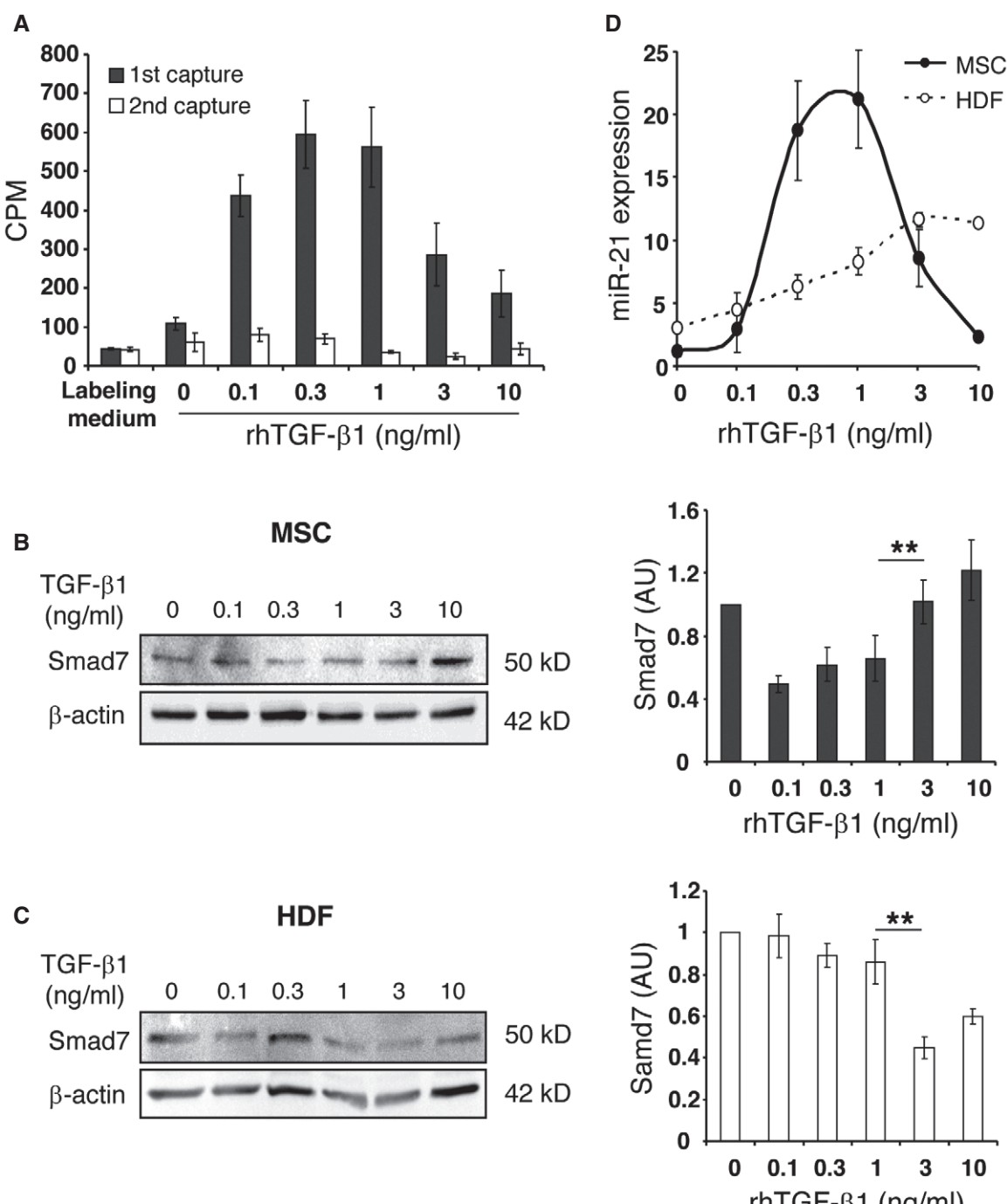

**Figure 6. TGF-β1 signaling in MSCs is regulated by environmental TGF-β1 concentration.**

A    AT-MSCs were cultured in the labeling media containing [$^{35}$S]-Met/Cys and increasing concentrations of r.h. TGF-β1 at 0, 0.1, 0.3, 1, 3, and 10 ng/ml for overnight. Supernatants were harvested and incubated with TGF-β1 capture antibody. After washing, the captured TGF-β1 was transferred to glass filter membrane, and the radioactivity was counted with a scintillation counter (black bars). The supernatants were incubated with new TGF-β1 capture antibody, and the radioactivity of captured TGF-β1 was counted again (white bars). The data are given as mean ± SD of five repeated measurements. This experiment was performed four times with comparable results. CPM, count per minute.

B, C  AT-MSCs (B) or HDFs (C) were cultured in media containing increasing concentrations of r.h. TGF-β1 at 0, 0.1, 0.3, 1, 3, and 10 ng/ml for 24 h, before protein lysates or total RNAs were harvested. Western blot analysis of Smad7 was performed with protein lysates and semi-quantitatively analyzed with densitometry. β-actin served as loading control. Data are given as mean ± SEM of three independent blots. **$P < 0.01$, by one-way ANOVA with Tukey's test. AU, arbitrary unit.

D    From total RNA, the mature miR-21 and endogenous control RNU6B were converted to cDNA and quantified by qPCR-based TaqMan microRNA assays. The expression of miR-21 was normalized on RNU6B. Data are given as mean ± SEM of three independent experiments.

Source data are available online for this figure.

**Table 1. Top differentially expressed miRNA.**

| miRNA ID | TGF-β1 conc. (ng/ml) | | | |
|---|---|---|---|---|
| | 0 | 0.1 | 1 | 10 |
| hsa-miR-21 | 1,000 | 1,385 | 5,816 | 1,157 |
| hsa-miR-142-3p | 1,000 | 3,797 | 3,850 | 0,362 |
| hsa-miR-22 | 1,000 | 2,799 | 3,568 | 0,561 |
| hsa-miR-376c | 1,000 | 2,151 | 2,523 | 0,719 |
| hsa-miR-19a | 1,000 | 1,925 | 2,035 | 0,644 |
| hsa-miR-423-5p | 1,000 | 0,420 | 0,467 | 1,231 |
| hsa-miR-155 | 1,000 | 0,336 | 0,488 | 1,227 |
| hsa-miR-425 | 1,000 | 1,338 | 1,526 | 0,395 |

MSCs in the presence of low TGF-β1 concentrations in the culture medium (0.1–1 ng/ml) when compared to control inhibitor-transfected MSCs. However, at higher TGF-β1 concentrations in the culture medium, TGF-β1 mRNA expression significantly increased in miR-21 inhibitor-transfected MSCs. These data suggest a high complexity in the regulation of environmental TGF-β1 sensing and suggest that at higher TGF-β1 concentrations in the culture media, miR-21 is induced in MSCs and cannot be fully neutralized by the miR-21 inhibitor.

To verify the role of miR-21 in MSCs *in vivo*, miR-21 inhibitor-transfected or control inhibitor-transfected MSCs were intradermally injected into CD18$^{-/-}$ wounds. Silencing of miR-21 in AT-MSCs did not affect transplantation efficiency, survival, or proliferation of MSCs in wounds (Fig EV3). CD18$^{-/-}$ wounds injected with miR-21-silenced MSCs showed significantly larger wound size compared to wounds received control inhibitor-transfected MSCs at days 3 and 6 post-wounding (Fig 7K), and showed less total TGF-β1 in the day 6 wounds (Fig 7L). These data demonstrate miR-21 is crucial for TGF-β1 production in MSCs in CD18$^{-/-}$ wounds, a microenvironment with critically low TGF-β1.

## Discussion

The main discovery of this report is that MSCs have evolved a previously unreported sensing mechanism for TGF-β1 levels at the wound site, and are endowed with the capacity to adaptively release TGF-β1 into wounds with reduced TGF-β1 as occurring in a murine LAD1 model closely reflecting human LAD1. Most importantly, the adaptive release of TGF-β1 from MSCs fully restored impaired wound healing in LAD1 mice. This MSC-specific sensing mechanism requires activation of the TGF-β receptor which depends on the TGF-β1 concentrations at the wound site, via changes in the miR-21 levels, either suppress or upregulate Smad7, and in consequence, adapt the release of MSC-derived TGF-β1 exactly to the demands at the wound site of LAD1 wounds (see graphical sketch Fig 8).

Accordingly, in LAD1 wounds that are used as a model for TGF-β1 deficiency in chronic wounds, activation of the TGF-β receptor induces miR-21, which blocks the translation of Smad7, thus relieves the suppression of TGF-β1 expression by Smad7, and in consequence, promotes enhanced TGF-β1 release from MSCs (Fig 8A). By contrast, overactivation of the TGF-β receptor in MSCs at high TGF-β1 concentrations down-regulates miR-21 and thus no longer inhibits Smad7 translation, eventually suppressing the release of endogenous TGF-β1 from MSCs (Fig 8B).

The observation that kinetics of TGF-β1 expression per MSC during wound healing are distinct and inversely correlated with environmental TGF-β1 concentrations at the wound site of WT and CD18$^{-/-}$ wounds (Fig 2D) suggests that MSCs can sense and shape their neighborhood. This notion is strongly supported by our finding that either overexpression or silencing of components of the TGF-β1 sensing and signaling pathway almost completely abrogates the adaptive response of MSCs. In this regard, (i) silencing TGF-β1 expression in MSCs, (ii) pharmacologically blocking the TGF-β receptor activity, (iii) silencing Smad7 expression, and (iv) overexpression or silencing miR-21 (Fig 8B) perturbed the adapted TGF-β1 release from MSCs in response to defined environmental conditions.

**Figure 7. Release of TGF-β1 by MSCs is regulated by miR-21/Smad7 signaling by sensing the environmental TGF-β1.**

A  AT-MSCs were pretreated with a TGF-βRI inhibitor SB431542 at 10 μM, or control DMSO for 1 h, and subsequently exposed to r.h. TGF-β1 at indicated concentrations. The cells were cultured for another 24 h and harvested for miR-21 assay. The expression of miR-21 was normalized on RNU6B. Data are given as mean ± SEM of three independent experiments.

B–D  AT-MSCs were transfected with 30 nM Smad7-siRNAs or control siRNA. Untransfected AT-MSCs served as control. RNA and protein were isolated 2 days after transfection for Smad7 expression by qPCR (B) and Western blot (C). The remaining cells were treated with r.h. TGF-β1 at indicated concentrations for another 24 h. The expression of human TGF-β1 was analyzed by qPCR and normalized on human GAPDH (D). Data are given as mean ± SEM of three independent experiments.

E–G  AT-MSCs were transfected with 5 nM miR-21 mimic or control mimic and were exposed to r.h. TGF-β1 2 days after transfection at indicated concentrations for another 24 h. The expression of miR-21 was analyzed by qPCR-based miR-21 assay normalized on RNU6B (E), Samd7 protein level was shown by Western blot (F), and human TGF-β1 expression was monitored by qPCR normalized on human GAPDH (G). Data are given as mean ± SEM of three independent experiments.

H–J  AT-MSCs were transfected with 50 nM miR-21 inhibitor or control inhibitor, and were exposed to r.h. TGF-β1 2 days after transfection at indicated concentrations for another 24 h, and subsequently subjected to the analysis of expression of miR-21 (H), Smad7 protein (I), and human TGF-β1 mRNA (J). Data are given as mean ± SEM of three independent experiments.

K, L  Full-thickness excisional wounds were produced on CD18$^{-/-}$ mice by 6-mm biopsy punches. One day after wounding, each wound received intradermal injection of 2.5 × 10$^5$ AT-MSCs that had been transfected with 50 nM miR-21 inhibitor or control inhibitor 2 days after transfection. CD18$^{-/-}$ wounds received mock injection with PBS served as control. Each wound region was digitally photographed at days 0, 3, and 6 post-wounding, and expressed as percentage of the initial wound size at day 0 (K). Wound tissues were harvested on day 6 post-wounding for quantification of total TGF-β1 protein by ELISA (L). Data are given as mean ± SEM; *n* = 5 wounds per group; *P < 0.05; **P < 0.01, by one-way ANOVA with Tukey's test. MSC_miR-21-IN, miR-21 inhibitor-transfected AT-MSCs; MSC_ctrl-IN, control inhibitor-transfected AT-MSCs.

Source data are available online for this figure.

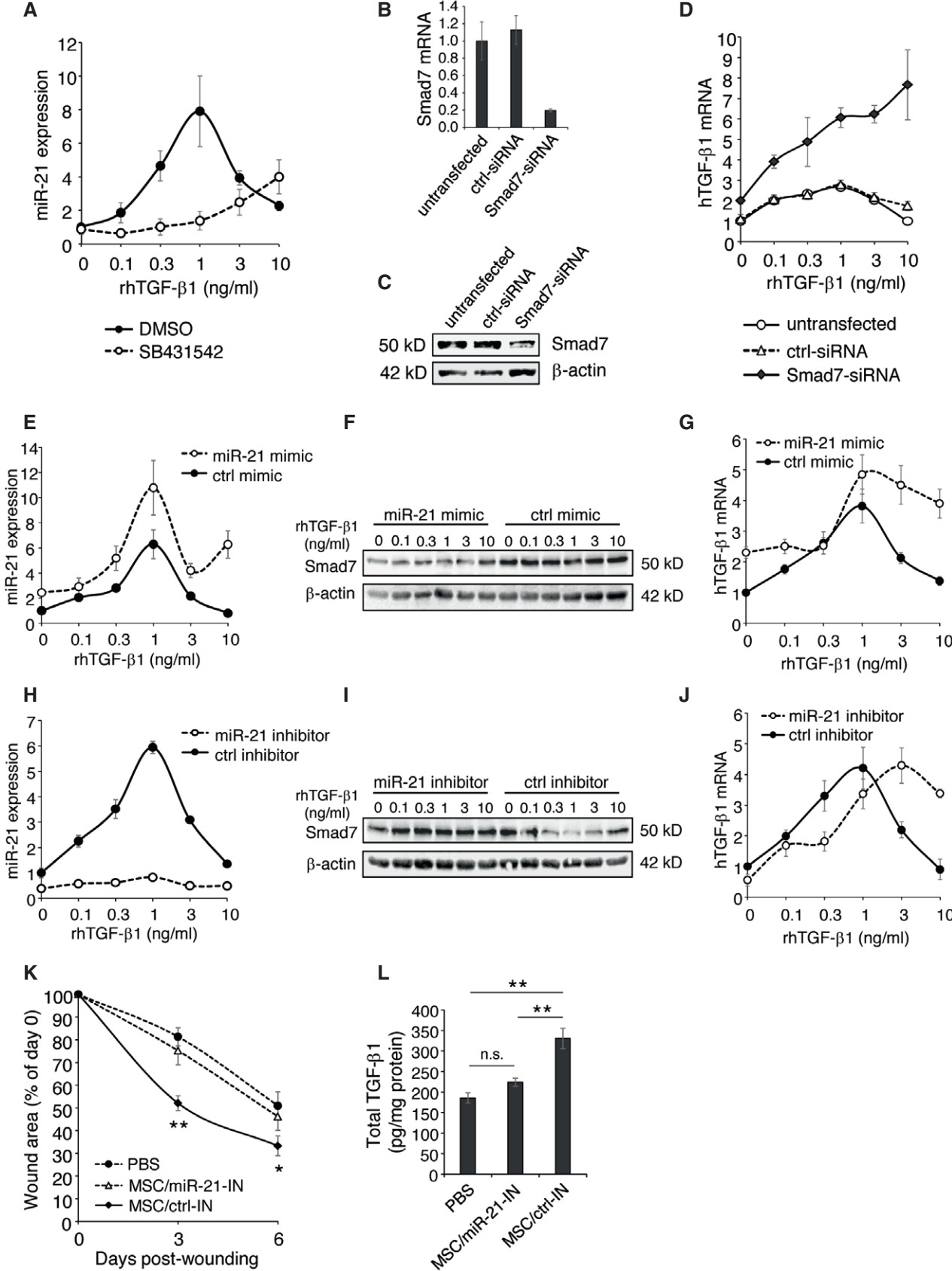

**Figure 7.**

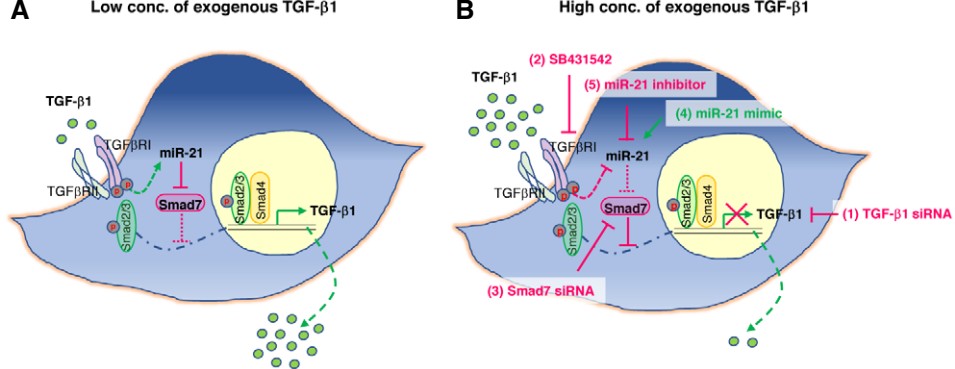

**Figure 8. Scheme of adaptive MSC responses to different environmental TGF-β1 levels.**

A  In the environment with low concentrations of TGF-β1, activation of TGF-β receptor induces miR-21, which blocks the translation of Smad7, thus relieves the suppression of TGF-β1 expression by Smad7, and promotes TGF-β1 release from MSCs.

B  Overactivation of TGF-β receptor in MSCs by high environmental TGF-β1 down-regulates miR-21 and thus no longer inhibits Smad7 translation and leads to close the TGF-β1 signal circuit. Such adaptive response is disturbed by deliberately changing the expression of components in the signaling pathway, such as (1) silencing TGF-β1 expression in MSCs by TGF-β1 siRNA, (2) blocking TGF-β receptor activity by chemical inhibitor SB431542, (3) silencing Smad7 expression by Smad7 siRNA, (4) overexpression of miR-21 by miR-21 mimic, and (5) knockdown miR-21 by miR-21 inhibitor.

It is widely accepted that excessive TGF-β1 drives fibrosis, a major cause of organ dysfunction, morbidity, and mortality [31]. Enhanced TGF-β1 expression is also related to the progression of several malignancies [32], and fibroproliferative conditions with contractures and restriction in joint mobility [33]. *Vice versa*, lack of TGF-β1 or reduced TGF-β1 levels as occurring in LAD1, chronic venous leg ulcers, and diabetic ulcers [3,34] results in severe pathology of non-healing wounds with a severe socioeconomic burden for societies. Therefore, an adapted control of TGF-β1 is essential for tissue homeostasis and tissue repair. In the light of these important morbidities related to dysregulated TGF-β1, our finding of TGF-β1 sensing and subsequently mounting an adaptive response in MSCs is particularly important as it is a unique and an unprecedented opportunity to control and coordinate TGF-β1 levels during wound healing and related disorders. In healthy human serum, the concentration of total TGF-β1 ranges from 30 to 50 ng/ml [35,36], and the concentration of active TGF-β1 is around 2–3 ng/ml [37]. In patients with tumors or inflammatory diseases, the TGF-β1 concentration is even higher [38]. Moreover, we cannot exclude the possibility that transplanted MSCs experienced higher TGF-β1 concentrations in their local microenvironment than the detected average measured in tissue homogenates. In fact, the focal release of TGF-β1 from the extracellular matrix by focal proteolysis or from cells in close vicinity can contribute to locally high TGF-β1 concentrations [39]. With these considerations, recombinant human TGF-β1 at a concentration range from 0.1 to 10 ng/ml was supplied *in vitro* to simulate the low and high environmental TGF-β1 in this study.

The skill to sense and shape their neighborhood is most likely enforced and optimized by evolution. Adaptation to environmental cues and maintenance of a proper niche environment constitute an essential advantage for the regulated self-renewal, quiescence, and differentiation of MSCs and is essential for tissue/organ homeostasis and repair. Importantly, niche homeostasis also protects from damage to the DNA as the critical blueprint harboring all the information for progenies originating from stem cells. In fact, dysregulated TGF-β1 concentrations affect these stem cell properties and

thus may drive a variety of pathological conditions [40]. Accumulating evidence suggests that the adaptive response of MSCs is important for niche homeostasis, yet the underlying mechanisms are largely unexplored. In this regard, MSCs suppress inflammatory cells including effector T cells [41], B cells [42], and *vice versa* enhance regulatory T cells that suppress effector T cells [43]. MSCs also promote the switch from an unrestrained activation of pro-inflammatory M1 macrophages to anti-inflammatory M2 macrophages [13,14]. Recently, we have reported that dermal resident MSCs in their endogenous niche or transplanted MSCs release soluble superoxide anion dismutase (SOD3) to counteract oxidative stress from overactivated neutrophils and reduce tissue damage [44]. Calprotectin (S100A8/A9), a danger-associated molecular pattern protein, is able to prime MSCs through TLR4 signaling for augmented wound repair capacity [45]. We now provide a clinically highly relevant example for the dissection of effector molecules responsible for adaptive TGF-β1 response mounted by MSCs in the interest of niche and tissue homeostasis. This is novel and bears the unprecedented opportunity to therapeutically exploit this mechanism for MSC-based therapies in a variety of TGF-β1 dysregulated conditions.

The function of miR-21 and Smad7 in TGF-β1 signaling has been documented in earlier studies. miR-21 is known to directly bind to the 3′UTR of Smad7 and suppress its translation [21–24]. Smad7 serves as the negative regulator of TGF-β1, by preventing association of Smad2/3 with TGF-β receptor [25], or targeting TGF-β receptor for degradation [46]. The upregulation of miR-21 results in suppression of the negative regulator Smad7, and in enhanced TGF-β1 expression and release. The upregulation of miR-21 in keloid scar fibroblasts sustains the positive feedback loop of TGF-β1 signaling that leads to uncontrolled proliferation of scar fibroblasts [47], likely due to the TGF-β1 autoinduction described three decades ago [48]. This is in line with our observation in HDFs that are unable to respond adaptively to environmental TGF-β1 concentrations. In contrast to fibrogenic fibroblasts, we here uncovered in MSCs that miR-21 plays an important role in sensing environmental TGF-β1

levels and relaying this information to Smad7. The miR-21/Smad7 signal may be responsible for the molecular control of fine tuning the balance of TGF-β1 autoinduction and TGF-β receptor degradation in response to environmental changes in TGF-β1 concentrations. Our observation is supported by recent studies showing upregulation of miR-21 in MSCs of various origins, and miR-21 is suggested to function as an essential component of an environmental and mechanical memory [49,50]. In addition, it has been shown that miR-21 can be packed in exosomes and released by MSCs to regulate TGF-β/Smad2 pathway and suppress fibrosis during wound healing [51]. The involvement of miR-21 and Smad7 in sensing environmental TGF-β1 levels in MSCs is of major relevance in rebalancing tissue and niche homeostasis during wound healing. In fact, miR-21 has been reported to be involved in different phases of cutaneous wound healing including granulation tissue formation, wound contraction, and fibrosis [52,53]. Similar as during wound healing, in mouse models of hind limb ischemia and xenograft tumor growth, where TGF-β is mandatory for angiogenesis, overexpression of miR-21 in transplanted AT-MSCs leads to increased blood flow recovery and induced tumor growth, respectively [54].

In aggregate, the here reported sensing response circuit in MSCs is novel and provides exciting unprecedented insight why MSC-based approaches may qualify as the preferred treatment superior to injection of recombinant TGF-β1 into non-healing wounds of patients suffering from LAD1 or other non-healing wounds. In fact, recombinant TGF-β1 is easily degraded in chronic human wounds [55,56], and repetitive injections of recombinant TGF-β1 into fibrotic wound edges so far have neither been successful nor established in clinical routine. Our data clearly underscore the advantage of a therapy with MSCs that rely on their sensing capacity and, in consequence, can be exploited as "adaptive drugstore" that depends on microenvironmental demands at the wound site substitute for TGF-β1 deficiency.

# Materials and Methods

### Mice

CD18-deficient (CD18$^{-/-}$) mice (B6.129S7-*Itgb2*$^{tm2Bay}$/J) and the corresponding wild-type (WT) control mice were housed and bred in specific pathogen-free facility at University of Ulm. Wound healing studies were performed using mice in experimental cohorts of the matched gender at an age of 8–12 weeks. Mice within a same strain (wild type or CD18$^{-/-}$) were randomly selected and allocated to either control or experimental groups. All experiments were carried out in compliance with the German Law for Welfare of Laboratory Animals and the ARRIVE reporting guidelines. The animal experiments were approved by the government of Baden-Württemberg with project numbers 1117 and 1396.

### Adipose tissue-derived mesenchymal stem cells

Human AT-MSCs at passage 2 were purchased from PromoCell (Heidelberg, Germany). AT-MSCs were seeded at a density of 3,000 cells/cm$^2$ in complete MSC growth medium with SupplementMix (PromoCell) and cultured at 37°C under 5% CO$_2$. AT-MSCs were harvested with Accutase (PAA Laboratories, Pasching,

Austria) at 70–80% confluence. AT-MSCs at passages 4–6 were used in the subsequent experiments. AT-MSCs express typical MSC markers including CD73, CD90, CD105, CD44, CD59, CD71, and CD29 and do not express leukocyte marker CD45, hematopoietic stem cell marker CD34, monocyte marker CD14, T cell marker CD3, or endothelial marker CD31. They are MHC class I positive but MHC class II negative (Fig EV5A). AT-MSCs in culture are plastic adherent (Fig EV5B) and are able to differentiate into adipocytes (Fig EV5C), osteoblasts (Fig EV5D), and chondrocytes (Fig EV5E and F) under respective induction media.

### Wound healing model

Prior to injury, CD18$^{-/-}$ or WT mice were anaesthetized by intraperitoneal injection of a ketamine (10 g/l)/xylazine (8 g/l) solution (10 μl/g body weight). After shaving the dorsal hair and cleaning the exposed skin with 70% ethanol, full-thickness (including the *panniculus carnosus*) excisional wounds were punched at two sites in the middle of the dorsum using 6-mm biopsy round knives (Stiefel, Offenbach, Germany). One day after wounding, AT-MSCs were harvested and washed with PBS for intradermal injections around the wounds. Each wound received three injections of 50 μl of AT-MSC suspension at density of $1.67 \times 10^6$ cells/ml, which led to $2.5 \times 10^5$ AT-MSCs per wound, and $1 \times 10^6$ AT-MSCs per mouse. In the control groups, each wounds received three mock injections of 50 μl PBS. Each wound region was digitally photographed at indicated time points, and wound areas were measured using Photoshop software (Adobe Systems, San Jose, CA). The researchers who performed imaging analysis were blinded to the mice grouping information. Wound sizes at any given time point after wounding were expressed as percentage of initial (day 0) wound area. Photography and wound area analyses were done in a blinded fashion.

### Flow cytometry

Fluorochrome (FITC or PE or APC)-conjugated anti-human antibodies CD73, CD90, CD105, CD71, CD29, CD31, CD34, CD45, HLA-ABC, and HLA-DR were purchased from eBioscience (CA, USA). AT-MSCs were harvested, washed with PBS, and incubated with antigen-specific antibodies for 30 min at room temperature. The non-specific staining was controlled by isotype-matched antibodies. Flow cytometry was performed on a FACSCanto II (BD Biosciences, CA, USA) with FACSDiva software for data acquisition (BD Biosciences). Data were analyzed with Summit software v4.3 (Beckman Coulter, CA, USA).

### Adipogenic, osteogenic, and chondrogenic differentiation of AT-MSCs

The induction of adipogenic, osteogenic, and chondrogenic differentiation of AT-MSCs and the subsequent Oil Red O staining, Alizarin Red S staining, and immunofluorescence staining of collagen type II and aggrecan were performed as described previously [13,44].

### Transfection of small RNAs

Adipose tissue-MSCs were seeded in antibiotic-free medium one night prior to transfection. The human TGF-β1 siRNA, human

Smad7 siRNA and AllStars negative control siRNA, human microRNA-21 (miR-21) mimic and control mimic, and human miR-21 inhibitor and control inhibitor were purchased from Qiagen (Hilden, Germany). The siRNA sequences were subjected to BLAST analysis to minimize the potential off-target effects. The sequences targeting human TGF-β1 are 5′-CAGCATATATATGTTCTTCAA-3′ and 5′-CACGTGGAGCTGTACVAGAAA-3′. The sequences targeting human Smad7 are 5′-CTGGATATCTTCTATGATCTA-3′ and 5′-CAGGCATTCCTCGGAAGTCAA-3′. AT-MSCs were transfected with siRNAs at final concentration of 30 nM, or with miRNA mimics at 5 nM, or miRNA inhibitors at 50 nM, respectively, by using Lipofectamine RNAiMAX (Invitrogen, Germany) diluted in Opti-MEM-reduced serum medium (Invitrogen). The transfection medium was replaced by MSC growth medium 6 h after transfection. The cells were harvested 24 h after transfection for intradermal injection or coculture with mouse fibroblasts, and 48 h after transfection for mRNA and protein expression analysis. For the treatment of transfected AT-MSCs with recombinant human TGF-β1, TGF-β1 was added at the respective concentrations 48 h after transfection and the cells were cultured for another 24 h.

**Real-time PCR and real-time reverse transcription PCR**

Genomic DNA was isolated from mice wounds at the indicated time points using the Easy-DNA Kit (Invitrogen). Total RNA was extracted with RNeasy Fibrous Tissue Kit (Qiagen), and cDNA was reverse transcribed with the First Strand cDNA Synthesis Kit (Fermentas, DE, USA) according to the manufacturer's procedures. Genomic DNA and RNA concentrations were measured by Nano-Drop (Thermo Scientific, DE, USA). Real-time PCR was performed in a volume of 10 μl containing QuantiTect SYBR Green PCR Master Mix (Qiagen), 0.5 μM forward and reverse primers, and 5 ng genomic DNA or cDNA using a LightCycler 480 II (Roche). Reactions were performed at 95°C for 15 min and followed by 50 cycles at 94°C for 15 s, 60°C for 20 s, and 72°C for 20 s. The sequences of the primers used in this study are summarized in Table 2. The human Alu assay was performed according to previously published procedures [57,58]. The standard curve was generated by real-time PCR of Alu sequence in DNA isolates from a mixture of a mouse wound and known number of AT-MSCs at $1 \times 10^2$, $1 \times 10^3$, $1 \times 10^4$, $1 \times 10^5$, and $5 \times 10^5$. The quantification of Alu sequences in mouse wounds was converted to the number of AT-MSCs in wounds based on the established standard curve. The relative expression of human GAPDH was normalized to mouse 18S rRNA.

**Radioactive labeling**

Adipose tissue-MSCs were seeded into six-well plate at the density of $2 \times 10^5$ cells/well in antibiotic-free culture medium containing DMEM, 10% FBS, 2% L-glutamine, and 25 mM HEPES. 24 h later, AT-MSCs were washed with antibiotic-free long-term labeling medium containing 90% Met/Cys-free DMEM (Life Technologies), 10% DMEM, 10% FBS, 2% L-glutamine, and 25 mM HEPES, treated with recombinant human TGF-β1 (PeproTech, NJ, USA) at various concentrations, and labeled with 0.05 mCi [35S]-Met/Cys (Hartmann Analytic, Braunschweig, Germany) in 1 ml of long-term labeling medium in 37°C, 5% CO₂ incubator overnight. The supernatants were harvested, treated with 1 N HCl followed by 1.2 N NaOH to

**Table 2. Primers for RT–PCR.**

| Target gene | Primer sequences |
|---|---|
| Mouse TGF-β1 | F: 5′-TGGAGCAACATGTGGAACTC-3′ |
| | R: 5′-GTCAGCAGCCGGTTACCA-3′ |
| Mouse 18S | F: 5′-GATCCCAGACTGGTTCCTGA-3′ |
| | R: 5′-GTCTAGACCGTTGGCCAGAA-3′ |
| Mouse β-actin | F: 5′-CCTTCTTGGGTATGGAATCCTGTGG-3′ |
| | R: 5′-CAGCACTGTGTTGGCATAGAGGTCTTTAC-3′ |
| Human TGF-β1 | F: 5′-GGACATCAACGGGTTCACTA-3′ |
| | R: 5′-GCCATGAGAAGCAGGAAAG-3′ |
| Human GAPDH | F: 5′-CGACCACTTTGTCAAGCTCA-3′ |
| | R: 5′-AGGGGTCTACATGGCAACTG-3′ |
| Human Alu | F: 5′-CATGGTGAAACCCCGTCTCTA-3′ |
| | R: 5′-GCCTCAGCCTCCCGAGTAG-3′ |
| Human β-actin | F: 5′-CACCACCGCCGAGACCGC-3′ |
| | R: 5′-GCTGGCCGGGCTTACCTG-3′ |
| Human Smad7 | F: 5′-CGATGGATTTTCTCAAACCAA-3′ |
| | R: 5′-ATTCGTTCCCCCTGTTTCA-3′ |

activate TGF-β1, and incubated in 96-well plates that had been pre-coated with anti- TGF-β1 capture antibody (R&D Systems) for 2 h with five technical repeats for each sample. After washing for three times with PBS, the bound proteins were collected and transferred to glass filter membrane, and the radioactivity was counted with a scintillation counter (PerkinElmer, MA, USA). After incubation, the supernatants from the first plate were collected and transferred to the second TGF-β1 capture antibody-coated plate for the second measurement.

**microRNA qPCR array**

$2 \times 10^5$ AT-MSCs were cultured alone or with 0.1, 1, or 10 ng/ml recombinant human TGF-β1 (PeproTech) for 24 h in duplicated wells. Thereafter, total RNA was purified by miRNeasy Kit (Qiagen) and 50 ng of total RNA from each sample was reverse transcribed to cDNA with miScript II RT Kit (Qiagen). cDNA was mixed with reagents from miScript SYBR Green PCR Kit (Qiagen) and used to profile the expression of 84 miRNA with miScript miRNA PCR Array Human miFinder (Qiagen, MIHS-001Z) in a 384-well plate format with a LightCycler 480 II (Roche) by following the manufacturer's instruction. The expression of miRNAs was normalized to the average of six housekeeping small RNAs (SNORD61, SNORD68, SNORD72, SNORD95, SNORD96A, and RNU6B) in individual samples. The average of duplicated samples was used to calculate the relative expression by using a conventional $2^{-\Delta\Delta Ct}$ method.

**microRNA assay**

Adipose tissue-MSCs or HDFs were treated with recombinant human TGF-β1 (PeproTech) at various concentrations for 24 h. The total RNA was extracted by RNeasy Plus Kit (Qiagen), and mature miRNA in 50 ng of total RNA was converted to cDNA by using TaqMan microRNA Reverse Transcription Kit (Life Technologies) together with the RT primer provided with the TaqMan MicroRNA

Assay. The expression of mature miR-21 was measured with TaqMan MicroRNA Assay (hsa-miR-21-5p, assay ID: 000397, Life Technologies) on ABI7300 (Life Technologies) by following manufacturer's instructions and normalized to endogenous control RNU6B (assay ID: 001093, Life Technologies).

### Blocking TGF-β receptor I (TGF-βRI)

SB431542 is a selective and potent inhibitor of activin receptor-like kinase (ALK)-4,5,7 (TGF-βRI) [28–30]. AT-MSCs were pretreated with 10 μM SB431542 (Tocris Bioscience, UK) or control DMSO for 1 h to inhibit the TGF-βRI, before the addition of recombinant human TGF-β1 at indicated concentrations. The cells were cultured for another 24 h and harvested for miR-21 assay.

### Coculture of AT-MSCs and mouse primary fibroblasts

Mouse fibroblasts were isolated from ears of 3- to 4-week-old C57BL/6 mice. Briefly, 70% ethanol sterilized minced ear was incubated in DMEM (Lonza) supplemented with 10% FBS, 2% L-glutamine, 1% penicillin/streptomycin, and 1% non-essential amino acids containing 2 mg/ml collagenase A (Roche) at 37°C for 2 h. Cell suspension was filtered through 70-μm mesh (Falcon). Cells were washed and seeded into culture medium and cultured in 37°C, 3% $O_2$, 5% $CO_2$ incubator. Fibroblasts were expanded for 3–4 passages before cocultured with human AT-MSCs, or TGF-β1/ control siRNA transfected AT-MSCs at the ratio of fibroblasts:AT-MSCs = 10:1. The coculture was kept at 37°C, 3% $O_2$ for 48 h. The culture of fibroblasts alone or AT-MSCs alone was served as controls. The culture of fibroblasts alone with 2 ng/ml recombinant human TGF-β1 was served as positive control for myofibroblast differentiation. Cells were either fixed with 4% paraformaldehyde (PFA) for immunofluorescence staining or lyzed with RIPA buffer (Sigma) supplemented with 2 mM $Na_3VO_4$, 10 mM NaF, and protease inhibitors (Roche) for Western blotting.

### Western blotting

Wound tissues or cultured cells were harvested at the indicated time points and homogenized in RIPA buffer (Sigma) supplemented with 2 mM $Na_3VO_4$, 10 mM NaF (Sigma), and protease inhibitors (Roche) with FastPrep-24 homogenizer in Lysing Matrix D tubes (MP Biomedicals, Germany). The homogenates were then centrifuged at 10,000 $g$ for 15 min at 4°C. The supernatant was collected and aliquoted, and stored in −80°C. The protein concentration was determined by Bradford assay (Bio-Rad). 50 μg protein from each sample was resolved by SDS–PAGE and transferred to nitrocellulose membranes (Whatman). Membranes were blocked with 5% BSA in TBS supplemented with 0.1% Tween-20 and incubated with primary antibodies against α-SMA (Clone ASM-1, Progen Biotechnik #61001), platelet endothelial cell adhesion molecule-1/CD31 (Novus Biologicals #NB100-2284), Smad2 (Clone D43B4, Cell Signaling Technology #5339), Phospho-Smad2 (Ser245/250/255, Cell Signaling Technology #3104), Smad3 (Clone C67H9, Cell Signaling Technology #9523), phosphor-Smad3 (Ser423/425, Clone C25A9, Cell Signaling Technology #9520; Cell Signaling Technology, MA, USA), Smad7 (MAB2029, R&D Systems #MAB2029), TGF beta receptor 2 (TGF-βRII; Santa Cruz #sc-400), or β-actin (Santa Cruz #sc-47778) overnight at 4°C. Thereafter, membranes were incubated with corresponding secondary antibody (anti-mouse IgG, anti-rabbit IgG, or anti-rat IgG) conjugated to HRP (Dianova, Germany) for 1 h at room temperature. Immunoreactions were detected by chemiluminescence using Vilber Fusion Fx7 system (Vilber Lourmat, Germany).

### Immunofluorescence

For formalin-fixed, paraffin-embedded wound tissues, 5-μm paraffin sections were rehydrated through 100, 96, 80, and 70% ethanol and finally distill water. Antigen retrieval was carried out by heating the sections in 1× target retrieval solution (DAKO) in steam for 15 min. For cryopreserved wound tissues or *in vitro* cultured cells, 5-μm cryosections of wounds or cultured cells in Lab-Tek II Chambered Coverglass (NUNC, Denmark) were fixed with 4% PFA. Slides were incubated with 5% BSA in PBS to block non-specific antibody binding and then incubated with primary antibodies at 4°C overnight, including anti-human beta-2 microglobulin (β2M; Abcam #ab15976), anti-α-SMA (Progen Biotechnik #61001), and anti- TGF-β1 (Acris Antibodies #AP06350PU-N). Rabbit IgG and mouse IgG were served as isotype controls, respectively. After wash with PBS for three times, the slides were incubated with fluorochrome-conjugated secondary antibodies (1:200, Invitrogen) for 1 h at room temperature, including Alexa Fluor 488-conjugated goat anti-rabbit IgG (for β2M), Alexa Fluor 555-conjugated goat anti-mouse IgG (for α-SMA), and Alexa Fluor 555-conjugated goat anti-rabbit IgG (for TGF-β1). Nuclei were counterstained with DAPI (Fluka). Coverslips were mounted with fluorescent mounting medium (DAKO). Photomicrographs were documented by using a Zeiss AxioPhot microscope with an AxioCam digital color camera and AxioVision software v4.7 (Zeiss, Germany).

The quantification of immunofluorescence staining was analyzed with ImageJ (ImageJ 1.51 [59]). The splitted single fluorescence channel in the duplicated image was converted to binary image with a fixed threshold value (threshold of α-SMA = 55, threshold of CD31 = 40). The area and mean gray value were measured by redirecting to the original image. The size of particle was set to a fixed range (100-infinity). The values of %Area in the summary were recorded.

### ELISA

The concentrations of TGF-β1 in the cell culture supernatants or wound lysates were measured by ELISA from R&D Systems (Minneapolis, MN, USA), by following the manufacturer's instructions.

### Statistical analysis

As stated in the corresponding figure legends, quantitative data are presented as mean ± SD or mean ± standard error (SEM). Two-tailed unpaired $t$-tests with Welch's correction or nonparametric Mann–Whitney tests were used to determine statistical significance between two groups; one-way ANOVA with Tukey's multiple comparison tests was used to determine the statistical significance in multiple groups.

**Expanded View** for this article is available online.

## Acknowledgements

This study is supported by research grants from the Baden-Württemberg Stiftung (P-BWS-ASII/15), the European Commission (CASCADE HEALTH-FP7-223236), the German Federal Ministry of Defence (E/U2AD/CF521/DF555), the German Research Foundation (SFB1149) to K.S.-K., and Baustein Program from the Medical Faculty, University of Ulm (LSBN.0100) to D.J.

## Author contributions

DJ and KS-K conceived the concepts and designed the experiments. DJ, KS, JM, SS, YQ, and EM conducted the experiments. DJ, AS, MW, and KS-K analyzed and interpreted the data. DJ and KS-K wrote the manuscript.

## Conflict of interest

The authors declare that they have no conflict of interest.

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
