## [Review Process File · EMBO Reports]

MSCs Rescue Impaired Wound Healing in a Murine LAD1 Model by Adaptive Responses to Low TGF- β 1 Levels

Dongsheng Jiang, Karmveer Singh, Jana Muschhammer, Susanne Schatz, Anca Sindrilaru, Evgenia Makrantonaki, Yu Qi, Meinhard Wlaschek, Karin Scharffetter-Kochanek

Review timeline:

Submission date:	21 August 2019
Editorial Decision:	22 November 2019
Revision received:	9 December 2019
Accepted:	31 January 2020

Transaction Report: This manuscript was transferred to *EMBO reports* following peer review at *EMBO Molecular Medicine*.

1st Editorial Decision

22 November 2019

Thank you for the submission of your research manuscript to our journal. The study has now been reviewed by former referee 1 who also had access to the referee reports from the other journal and your point-by-point response. As you can see from the comments below, the referee finds the revised study now suitable for publication in EMBO reports.

We therefore invite you to revise your study for publication in EMBO reports and to address the following editorial points that we need before we can proceed with the official acceptance of your study.

- 1) Please submit your manuscript as a .docx formatted version that contains only the text (including legends for main figures, EV figures and tables).
- 2) Please shorten the title to 100 characters including spaces and the abstract to 150 words.
- 3) We need individual production quality figure files as .eps, .tif, .jpg (one file per figure). Please also see our Figure Preparation Guidelines (figure preparation pdf) from our Author Guidelines pages <https://www.embopress.org/page/journal/14693178/authorguide> for more info on how to prepare your figures.
- 4) Supplementary Information: you currently have 6 Supplementary figures. You have two options:
 - You could combine them to 5 figures and submit them as Expanded View (EV) Figures. These are collapsible/expandable online. Table S1 could also be displayed in Expanded View. If you choose this option, we need the EV Figures as individual production quality files and the EV Figure legends should be included in the main text after the legends of regular figures. The nomenclature is "Figure EVx", "Table EVx".
 - Alternatively, you can combine all Supplementary figures/tables and their legends into a single pdf called "Appendix". The nomenclature for these is "Appendix Figure Sx", "Appendix Table Sx".

The Appendix needs a title page with a table of content including page numbers.

See detailed instructions regarding expanded view here:

[<https://www.embopress.org/page/journal/14693178/authorguide#expandedview>](https://www.embopress.org/page/journal/14693178/authorguide#expandedview)

5) Figure legends and statistics:

All figure legends must contain the following information:

- Graphs must include a description of the bars and the error bars (SEM, SD).
- The statistical test used to generate the error bars and P-values must be stated.
- The number of independent biological replicates underlying each data point must be given.

IMPORTANT: please note that error bars and statistical comparison may only be applied if the data are based on at least 3 independent biological replicates. If your data does not meet these criteria, either provide more samples or show the data as scatter blots.

- All microscopy images must contain a scale bar that is defined in the legend.

6) Please complete and upload the author checklist, which you can download from our author guidelines ([<https://www.embopress.org/page/journal/14693178/authorguide>](https://www.embopress.org/page/journal/14693178/authorguide)). Please insert information in the checklist that is also reflected in the manuscript. The completed author checklist will also be part of the RPF.

7) Please note that all corresponding authors are required to supply an ORCID ID for their name upon submission of a revised manuscript ([<https://orcid.org/>](https://orcid.org/)). Please find instructions on how to link your ORCID ID to your account in our manuscript tracking system in our Author guidelines ([<https://www.embopress.org/page/journal/14693178/authorguide#authorshipguidelines>](https://www.embopress.org/page/journal/14693178/authorguide#authorshipguidelines))

8) EMBO reports papers are accompanied online by A) a short (1-2 sentences) summary of the findings and their significance, B) 2-3 bullet points highlighting key results and C) a synopsis image that is 550x200-400 pixels large (width x height). You can either show a model or key data in the synopsis image. Please note that the size is rather small and that text needs to be readable at the final size. Please send us this information along with the revised manuscript.

9) We would also encourage you to include the source data for figure panels that show essential data. Numerical data should be provided as individual .xls or .csv files (including a tab describing the data). For blots or microscopy, uncropped images should be submitted (using a zip archive if multiple images need to be supplied for one panel). Additional information on source data and instruction on how to label the files are available

[<https://www.embopress.org/page/journal/14693178/authorguide#sourcedata>](https://www.embopress.org/page/journal/14693178/authorguide#sourcedata).

10) Our journal encourages inclusion of *data citations in the reference list* to directly cite datasets that were re-used and obtained from public databases. Data citations in the article text are distinct from normal bibliographical citations and should directly link to the database records from which the data can be accessed. In the main text, data citations are formatted as follows: "Data ref: Smith et al, 2001" or "Data ref: NCBI Sequence Read Archive PRJNA342805, 2017". In the Reference list, data citations must be labeled with "[DATASET]". A data reference must provide the database name, accession number/identifiers and a resolvable link to the landing page from which the data can be accessed at the end of the reference. Further instructions are available at [<https://www.embopress.org/page/journal/14693178/authorguide#referencesformat>](https://www.embopress.org/page/journal/14693178/authorguide#referencesformat).

11) As part of the EMBO publication's Transparent Editorial Process, EMBO reports publishes online a Review Process File to accompany accepted manuscripts. This File will be published in conjunction with your paper and will include the referee reports, your point-by-point response and all pertinent correspondence relating to the manuscript.

We would also welcome the submission of cover suggestions, or motifs to be used by our Graphics

Illustrator in designing a cover.

I look forward to seeing a revised version of your manuscript when it is ready. Please let me know if you have questions or comments regarding the revision.

REFEREE REPORTS

Referee #1:

The study by Jiang and coworkers has previously been submitted to EMBO Molecular Medicine and was reviewed by this reviewer. The authors addressed all technical concerns either with new experiments and data or by down-toning interpretations that were not well substantiated. They have included essential references in their discussion and experimental considerations.

One major initial concern was the lack of conceptual novelty of the first submission considering the substantial work already published on the roles of TGF and miR-21 in a wound healing context. However, the authors now better support and discuss that MSCs delivered to a wound bed can adapt their levels of TGF production to the local environment. Their data support that miR-21 and Smad7 are involved in this rheostat functions and that this function is differently regulated in MSCs compared with dermal fibroblasts. The manuscript has been restructured and writing has been improved which clarified most if not all initial uncertainties.

In summary, the initial submission the work has been very much improved and is consistent with the standards requested by EMBO Reports. This reviewer has no further concerns.

1st Revision - authors' response

9 December 2019

The authors performed all minor editorial changes.

Corresponding Author Name: Karin Scharffetter-Kochanek

Manuscript Number: EMBOR-2019-49115V1